# Sulfonamides with Heterocyclic Periphery as Antiviral Agents

**DOI:** 10.3390/molecules28010051

**Published:** 2022-12-21

**Authors:** Mikhail Yu. Moskalik

**Affiliations:** Irkutsk Institute of Chemistry, Siberian Branch of the Russian Academy of Sciences, 1 Favorsky Street, 664033 Irkutsk, Russia; moskalik@irioch.irk.ru

**Keywords:** sulfonamide, *N*-heterocycles, antiviral activity, multi-step syntheses, piperidine, morpholine, azoles, azines

## Abstract

Sulfonamides are the basic motifs for a whole generation of drugs from a large group of antibiotics. Currently, research in the field of the new sulfonamide synthesis has received a “second wind”, due to the increase in the synthetic capabilities of organic chemistry and the study of their medical and biological properties of a wide spectrum of biological activity. New reagents and new reactions make it possible to significantly increase the number of compounds with a sulfonamide fragment in combination with other important pharmacophore groups, such as, for example, a wide class of *N*-containing heterocycles. The result of these synthetic possibilities is the extension of the activity spectrum—along with antibacterial activity, many of them exhibit other types of biological activity. Antiviral activity is also observed in a wide range of sulfonamide derivatives. This review provides examples of the synthesis of sulfonamide compounds with antiviral properties that can be used to develop drugs against coxsackievirus B, enteroviruses, encephalomyocarditis viruses, adenoviruses, human parainfluenza viruses, Ebola virus, Marburg virus, SARS-CoV-2, HIV and others. Since over the past three years, viral infections have become a special problem for public health throughout the world, the development of new broad-spectrum antiviral drugs is an extremely important task for synthetic organic and medicinal chemistry. Sulfonamides can be both sources of nitrogen for building a nitrogen-containing heterocyclic core and the side chain substituents of a biologically active substance. The formation of the sulfonamide group is often achieved by the reaction of the *N*-nucleophilic center in the substrate molecule with the corresponding sulfonylchloride. Another approach involves the use of sulfonamides as the reagents for building a nitrogen-containing framework.

## 1. Introduction

*N*-heterocycles and linear products synthesized from sulfonamides or containing a sulfonamide fragment in the side chain are important objects for studying biological activity. Traditionally, sulfonamides are primarily considered as antibacterial agents; however, among these substances one can find not only effective antibiotics [1,2], but also compounds with very different activities: oral hypoglycemic [3,4], antitumor [3,5], antiviral [6,7,8,9], antiepileptic [10], antihypertensive [11], antiprotozoal [12], antifungal [13], anticancer [14,15,16], anti-inflammatory [17], diuretic [18], butyrylcholinesterase inhibitors (anti-Alzheimer’s disease activity) [19], MAO-B specific inhibitors (activity in the treatment of neurodegenerative disorders such as Parkinson’s disease) [20], COX-2 inhibitors (therapy of inflammatory disease) [21], etc. Recent advances in the medicinal chemistry of sulfonamides showed the possibilities of new unique drug design. Sulfonamides are of particular importance in the synthesis of carbonic anhydrase inhibitors, which are used, for example, in combined chemotherapy for cancer [22]. Heterocyclic sulfonamides acts as hCA IV (as drug targets) inhibitors [23]. Pyrrolidine-containing sulfonamides (hCA IV inhibitors) are promising drugs for the treatment of glioma [23]. Sulfonamide hCA VII inhibitors are used in the complex therapy of HIV-infection. The *N*-acylsulfonamide form of the prodrug Elsulfavirin selectively inhibits hCA VII to treat neurological complications of HIV infection [24]. Sulfonamide-containing 1,3-oxazoles and thiophenes show cytosolic CA I and CA II inhibition in the picomolar concentrations with extremely high selectivity [25]. Obtained compounds can currently be considered as the most potential basic structures of CA inhibitors for the synthesis of a wide range of drugs [25]. The authors of [26] obtained next-generation sulfonamide-containing carbonic anhydrase inhibitors that demonstrated high and selective inhibition effect of glaucoma-related hCA II, high hydrophilicity and the possibility of conjugation to sustained-release nanoparticles. Even at the initial stage of research, 5-(sulfamoyl)thien-2-yl 1,3-oxazoles (eye drops, 1% solution) showed an intraocular pressure reduction similar to the clinically used 2% solution of dorzolamide [26]. Sulfonamide derivatives often do not require complex functionalization of substituents [27], that greatly facilitates the synthesis of compounds used in the treatment of a wide range of diseases. It is widely known that nitrogen-containing heterocycles are the part of a large number of drugs of synthetic and natural origin [28,29]. Combining sulfonamide and *N*-heterocyclic pharmacophore groups in a molecule is one of the effective approaches to the synthesis of broad-spectrum drugs.

## 2. Antiviral Sulfonamide Derivatives

In their work, [30] presented the synthesis of chiral *N*-heterocycles based on arylsulfonamides. 2-Azabicyloalkane derivatives (2-azabicyclo[2.2.1]heptane and 2-azabicyclo[3.2.1]octane) contained dansyl- and biphenylsulfonamide fragments (compounds **3** and **6**). The resulting compounds exhibited antiviral activity against HPIV-3 and EMCV (Figure 1):

The reactions were diastereoselective and enantioselective. Compounds **3** and **6** were obtained by the classical method for sulfonamide chemistry, which consists in the treatment of amines (in this case, bicyclic amines **2** or **5**) with available aryl-substituted sulfonyl chlorides (**1** or **4**). Substrates **2** and **5** were obtained by a stereoselective aza-Diels-Alder reaction (cyclopentadiene with in situ generated Shiff base reaction [30,31]). The most significant antiviral activity of compounds **3** and **6** was observed against EMCV with IC_50_ = 22.0 ± 2.6 µM and IC_50_ = 18.3 ± 2.0 µM, respectively, with selectivity indexes (SI) of 40.3 and 19.6. Lower activity of 2-azabicyclo[2.2.1]heptanesulfonamide **6** was also observed against AdV5 and HPIV-3 with IC_50_ = 7.5 ± 0.8 μM (SI = 1.8) and IC_50_ = 1.5 ± 0.2 μM (SI = 2.8). At the same time, 2-azabiycolo[3.2.1]octane **3** showed minor antiviral activity against HPIV-3 [30].

Camphor derivatives containing heterocyclic fragments of the sulfonamide nitrogen atom were obtained in a similar way. The compounds exhibit antiviral activity against filoviruses (Ebola and Marburg viruses) [32] (Figure 2):

Compound **8** was obtained by the reaction of camphor-substituted sulfonyl chlorides with the corresponding *N*-nucleophiles in the presence of Et_3_N in CH_2_Cl_2_. In the study of biological activity, sertraline was used as a reference drug, which has shown its efficacy in EBOV therapy. The resulting sulfonamide, **8**, containing morpholine and triazole fragments, exhibited inhibitory activity against EBOV glycoproteins comparable to that of sertraline. Thus, as suggested by the authors of [17], the approach based on the synthesis of *N*-heterocyclic sulfonamides containing bicyclic terpenoid, camphor or borneol moieties is promising for the inhibitors research against dangerous viral infections [32]. Analysis of the inhibitory activity of **8** against glycoproteins showed that the minimum IC_50_ value was discovered for piperidine-substituted sulfonamides. This information is consistent with the results published earlier in [33], where a borneol derivative containing an ester group and a piperidine fragment was obtained and characterized. The results showed a good biological activity inhibiting MARV intrusion into the cell [33].

Morpholine-substituted sulfonamide **17** was synthesized, which exhibited biological activity against avian paramyxovirus (APMV-1) [34] (Figure 3):

When tested, sulfonamide **17** was shown to exhibit three times higher antiviral activity against APMV-1 than ribavirin, a commonly available antiviral drug [34]. The multi-step synthesis of compound **17** is shown in the Figure 3. 3,3-Dimethylcyclohexanone **9** was chosen as the starting substrate for the synthesis of sulfonamide **17**; its treatment with dimethyl carbonate in the presence of sodium hydride in THF gave carboxylate **10**. Compound **10** was heterocyclized in the presence of S-methylisothiourea in water followed by treatment with KOH to form tetrahydroquinazoline-4 **11** (these steps of the synthesis are described in [35,36]). The intermediate *N*-heterocycle **11** was further reacted with morpholine 12 on heating to 120 °C to give product **13** [37]. The synthesis of the target product 17 was completed by the substitution of the hydroxyl group for a chlorine atom in the presence of POCl_3_ with the formation of product **14**. Then the chlorine atom was substituted by piperazine to form compound **15** [38]. Compound **15** was further treated with sulfonylchloride **16** to give target product **17** [34].

Cyclic sulfonamide **22** (Figure 4) described in [39] exhibits a specific type of biological activity against SARS-CoV-2. Compound **22** was a potential inhibitor of SARS-CoV-2, did not exhibit cytotoxicity, had an IC_50_ = 0.8 µM and SI = 30.7. Heterocycle **22** showed good oral bioavailability (77%), metabolic stability and low binding to hERG. The study presented in [39] showed that heterocyclic sulfonamide **22** was a promising substance for the development of drugs—specifically, SARS-CoV-2 inhibitors [39]:

Sulfonyl chloride **18** was selected as the starting substrate for the product **22** synthesis. Compound **18** was added to aqueous ammonia and refluxed for 1 h to form sulfonamide **19**. The methyl group in **19** was oxidized with KMnO_4_ in a basic medium to form compound **20**, which was dehydrated to saccharin 21 under the action of sulfuric acid. Next, compound **21** was heterocyclized and functionalized in a three-step sequence to form product **22**, as shown in Figure 4 [39].

Hydantoin derivatives are important biologically active substances, many of which are well studied and have been used for several decades to obtain drugs [40,41]. Hydantoins are often considered as *N*-heterocycles containing an α-amino acid fragment and urea, which causes the presence of various types of biological activity, that can be easily changed by varying substituents [42,43]. Hydantoin-substituted sulfonamides showed antiviral activity [44,45] (Figure 5):

Product **28** was synthesized in two successive steps. At the first step, the amino group of polyfunctional amine **24** was attached to isocyanate **23**. Intermediate adduct **25** underwent heterocyclization to compound **26** due to the presence of a CN-group in the structure. Then, intermediate **26** was recyclized to the target intermediate **27**. Sulfonic chloride **27** was further treated with an excess of aliphatic primary or secondary amines to form *N*-heterocyclic sulfonamide **28**. Compound **28** was effective against cytomegalovirus strain AD169. The EC_50_ value of the sulfonamide **28** was comparable to the EC_50_ of the reference drugs (ganciclovir and cidofovir) [44,45].

Purine derivatives are rightly considered to be special structures for the study of properties in the field of medical and synthetic organic chemistry due to the wide presence of such fragments in natural compounds. Purines and fused purines are going to get a lot of attention for many years due to their interesting pharmacological properties in antiviral [46,47,48,49,50,51] and antimicrobial [52] drugs. By a simple reaction of 6-chloro-4,5-dihydro-7H-purine **29** with sulfamide derivative **30**, a new compound **31** exhibiting biological activity was obtained. Such compounds are very promising for treating various viral infections [46] (Figure 6):

The yields of compounds were about 80%.

Pyrrole rings are part of porphyrins, hemoglobins and cytochromes. Compounds containing pyrrole and sulfonamide fragments could potentially exhibit various types of biological activity. As part of the study of methods for the synthesis of antiviral drugs based on sulfonamides, it is worth noting the unique reaction presented in Figure 7. This reaction is a simple and effective method for obtaining substituted pyrrole **34** from primary sulfonamide **33** [53]:

The yields of the compounds were quantitative. The availability and low cost of the process is important for medicinal chemistry and the synthesis of biologically active nitrogen-containing heterocycles, given the presence of pharmacophore sulfonamide groups in the products. The reaction is applicable to a wide range of substrates, including arylsulfonamides, alkylsulfonamides **33** and various drug molecules containing sulfonamides. The formation of sulfonylpyrrole **34** proceeded via the reaction of sulfonamide and 2,5-dimethoxytetrahydrofuran (diMeOTHF) **32** in the presence of a catalytic amount of p-TsOH (similar to the reaction of sulfonylpyridinium salts [54]). The reaction was also carried out with microwave activation without solvent and additives at 150 °C [53]. The method may be of interest for the synthesis of antiviral drugs, since antiviral biological activity is known for sulfonamide-containing pyrroles [55].

The synthesis of small-molecule mimics involved in the interaction of the broadly neutralizing antibody 447-52D with the gp120 V3 loop was developed [56] (Figure 8). The resulting 1,3,5-triazine framework **39** showed very good antiviral activity. The molecules have an IC_50_ below 5.0 μM. The study was revealed a promising molecular structure that can be further studied to produce powerful HIV-1 inhibitors aimed at virus entry.

In the work [56] was presented the synthesis of 1,3,5-triazine-substituted sulfonamide **39**. The compound **39** showed good antiviral activity and had an IC_50_ below 5.0 μM. Product **39** was the basic molecular structure that can be used to synthesize inhibitors of HIV-1 virus entry into the cell (Figure 8):

The synthesis of compound **39** was carried out by successive treatment of cyanuric chloride **35** with sulfanilamide and the corresponding amines under alkaline conditions with cooling to 0–4 °C or at room temperature in acetone or dioxane. The reaction proceeded by the mechanism of nucleophilic substitution. The product was isolated with a good yield [56].

A similar synthesis methodology based on a 1,3,5-triazine derivative made it possible to obtain sulfonamides **45** (Figure 9) that exhibited high antiviral activity. Compound **45** [57] was up to six times more effective than ribavirin (RBV) against DENV (Dengue virus) and up three times more effective than 7-deaza-2′-C-methyladenosine (7DMA) [58] against ZIKV (Zika virus) (SI > 46 and > 41 respectively):

The pyrimidine compound represented the best candidate to develop broad-spectrum antiflavivirus agents after a focused optimization to further increase its potency and ADME properties [57].

Thiosemicarbazones are widely used substrates in the synthesis of substituted *N*-heterocycles and antiviral compounds with a wide spectrum of antiviral activity against a number of DNA and RNA viruses. Derivatives of thiosemicarbazones are known to exhibit inhibitors of RNA replication of hepatitis C virus and HIV-1 [59,60,61]. There were known sulfonamide derivatives of thiosemicarbazone **47** with antiviral properties, which had fewer side effects, rapid clearance rate or less incidence of relapse [62] (Figure 10):

The expected anti-BVDV (bovine viral diarrhea virus) property of the synthesized derivatives was tested. The results indicated that the presence of sulphonamido- at arylazo and ethyl-, phenyl- at *N*-(4)-thiosemicarbazone moieties exhibited a potent anti-BVDV activity [62].

In the treatment of severe infections such as HIV-1, combined antiretroviral therapy, including non-nucleoside reverse transcriptase inhibitors is used. This approach has shown to be effective in suppressing viral replication. There are sulfonamide-containing *N*-heterocycles, which have shown their effectiveness in antiretroviral therapy. Compound **55** was an inhibitor of the RT-enzyme in nanomolar concentrations, and inhibitor of HIV-1 replication in MT4 cells with minimal cytotoxicity [63] (Figure 11):

Compound **55** exhibited HIV-1 inhibitory activity in some cases with greater efficiency than the reference drugs (nevirapine and efavirenz) [63]. The multistage synthesis of product **55** started with the reaction of 2-nitroaniline **48** with 3,5-dimethylbenzenesulfonyl chloride **49** in DMF and in the presence of sodium hydride as a catalyst. In the resulting product **50**, the nitro group was further reduced to form the corresponding *N*-(2-aminophenyl)-3,5-dimethylbenzenesulfonamide **51** according to the classical method in the presence of zinc and hydrochloric acid in ethanol. In the next step, compound **51** was heterocyclized in the presence of 1,1-thiocarbonyldiimidazole **52** to give 1-(3,5-dimethylphenylsulfonyl)-1,3-dihydro-2H-benzimidazole-2-thione **53**. At the last stage, the intermediate heterocycle **53** was dissolved in DMF and treated with *N*-phenylacetamide **54** in the presence of K_2_CO_3_ to give benzimidazole **55**. The last reaction was carried out at room temperature overnight [63].

In the work [64] was presented a simple methodology for the synthesis of quinoline-substituted sulfonamide **60**. The resulting compound **60** was tested against four viruses that infect poultry. Analysis of antiviral activity and IC_50_ values showed that sulfonamide **60** is active against Newcastle disease virus (NDV), infectious bursal disease virus (IBDV), avian influenza virus subtype H9N2 (AIV) and infectious bronchitis virus (IBV) (Figure 12):

The first step in the synthesis of sulfonamide **60** was the Knoevenagel condensation of barbituric acid and aldehyde **57**, which proceeded with the formation of 5-(3-nitrobenzylidene)pyrimidine-2,4,6(1H,3H,5H)-trione **58**. Compound **58** underwent further heterocyclization with various sulfonamides **59** to form quinoline-substituted sulfonamide **60**. The sulfanonamide derivative **60** showed high activity against influenza viruses H9N2, NDV and IBDV. The thiazole derivative **60** was active against NDV and IBDV strains. The lowest IC_50_ (0.001 mg) was obtained in a study of biological activity against the H9N2 virus. The oxazole derivative **60** was active against all viruses except H9N2; the lowest IC_50_ value (0.01 mg) was shown for IBV. Guanidine-substituted sulfonamide **60** demonstrated the highest antiviral activity comparable to that of amantadine [64,65].

Calixarenes are cyclic oligomers synthesized from phenol and formaldehyde [66], that have been proposed as potential drug candidates [67,68]. Sulfanilamide derivatives bounded to the calix[4]arene **61** scaffold also showed potential antiviral activity [69]. It was reported that the introduction of the azo-group has improved more than 60% of the antibacterial activities of certain molecules [70]. At the moment, theoretical studies of biological activity have shown a high potential of as neuraminidase receptor inhibitors. The presence of an aza-group in the compound **62** suggested the possibility of synthesis of a number of nitrogen-containing heterocycles exhibiting antiviral activity [69]. The yield of compounds **62** (R = H- and guanidyl-) were 62 and 55 % (Figure 13):

The authors of [71] presented the synthesis of a new class of functionalized pyridines **65**, including sulfonamide fragments, exhibiting antiviral activity (Figure 14):

Compound **65** was obtained by the reaction of *N*-cyanoacetoarylsulfonylhydrazide **63** and 2-cyano-3-ethoxyacrylate **64** in the presence of Na alkoxide. Target compound **65** was formed via the addition of the active methylene group of *N*-cyanoacetoarylsulfonylhydrazide **63** to the C=C bond in 2-cyano-3-ethoxyacrylate **64**, followed by elimination of the ethanol molecule and cyclization due to the addition of the NH group to the nitrile group. The antiviral activity of new derivative **65** was determined in vitro against a wide range of viruses (herpes simplex virus type 1 (HSV-1), Coxsackie virus B4 (CBV4), hepatitis A virus HM 175 (HAV HM 175), ED.-43/SG-Feo (VYG), hepatitis C virus genotype 4a (HCVcc) and adenovirus type 7 (HAdV7)). For most of these viruses no specific drugs were found. Known drugs were used only to treat the symptoms of the disease (an exception is acyclovir) [72,73,74]. The compounds showed an impressive antiviral effect against three of the studied viruses (HSV-1, CBV4 and HAV) [71].

A new class of functionalized benzothiazole **71** bearing *N*-sulfonamide 2-pyridone derivatives was synthesized and its antiviral potency was exhibited [75]:

Arylsulfonohydrazide **68** was used as the starting compound for the synthesis of *N*-arylsulfonylpyridone **71**, that includes a benzothiazole fragment. The reaction of benzothiazole acetate **66** with hydrazine hydrate at room temperature gave acetohydrazide **67**. Further, on treatment of acetohydrazide **67** with arylsulfonyl chloride in pyridine at room temperature, product **68** was obtained as a result of sulfonation in high yield. Reaction of *N*-arylsulfonohydrazide **68** with sodium salt of 2-(hydroxymethylene)-1-cycloalkanone **69** in the presence of piperidine acetate led to the formation of target compound **71** through the formation of intermediate **70** in good yield (Figure 15) [75]. Antiviral studies in vitro against HSV-1, HAV HM175, HCVcc genotype, CBV4 and HAdV7 showed that the compound **71** exhibited good antiviral activity. The CC_50_ and IC_50_ values of the compound **71** were measured and their SI determined. In silico studies have shown that compound **71** has good oral bioavailability and can easily enter a cell [75].

Pyrimidine derivatives are important *N*-containing heterocycles that are widely involved in the biological processes occurring in the cell. Among them there are such compounds as uracil, cytosine, thymine. Pyrimidines are also found in many nucleotides, vitamins, coenzymes and antiviral drugs. A number of new substituted 2-pyrimidylbenzothiazoles containing sulfonamide fragments and exhibiting antiviral activity were synthesized [76] (Figure 16):

Pyrimidine **76** was synthesized by the reaction of *N*-arylsulfonylguanidine 75 with benzothiazole derivative **74** in the presence of alkali in dioxane in the absence of air. Intermediate alkene **74** was formed by the Michael reaction of nitrile **72** and aldehyde **73**. Antiviral activity against HSV-1, CBV4, HAV HM 175, HCVcc genotype 4 and HAdV7 viruses was studied for compound 76 [76]. In the case of HSV-1, the compound was found to exhibit excellent viral load reduction in the range of 70–90% with good IC_50_, CC_50_ and SI values compared to acyclovir. In the case of CBV4, a reduction in viral activity of more than 50% was shown. Compound **76** also had inhibitory activity against the Hsp90α protein with an IC_50_ in the range of 4.8–10.4 µg/mL [76].

Derivatives of 4-(1,3-dioxo-2,3-dihydro-1H-isoindol-2-yl)benzene-1-sulfonamide **80** were synthesized, which exhibit inhibitory activity against the Dengue virus. Compound **80** inhibited the NS2B-NS3 protease, which acts as a target for the development of anti-DENV2 agents. The resulting sulfonamide **80** showed IC_50_ values for DENV2 protease inhibition of 48.2 and 121.9 µM, respectively [77]. Compound **80** was obtained as shown in Figure 17: 

Sulphonyl chloride intermediate **79** was synthesized from dione **78** by chlorosulphonation reaction. The next step involved substitution of chloride in the intermediate with amino group in the presence of base. The yield of the reaction was 88% [77]. 4-(Phenylsulphonamido)benzoic derivative of **80** showed broad-spectrum activity against coxsackievirus B (CVBs), enteroviruses (EV) of groups C and D and even rhinoviruses (RV) [78]. Chemical modifications of benzoic derivative **80** led to the formation of compounds with high activity against the coxsackievirus B3 (Nancy, CVB3) strain (IC50 value of 4.29 µM [79] (Figure 18):

The phthalimide motif in compound **81** was hydrolyzed to form intermediate **82**. Ester **82** was also hydrolyzed with refluxing in the presence NaOH to form compound **83** (yield 82%) [79].

## 3. Conclusions

Viral infections have attracted the close attention of the entire scientific community in every corner of the planet. The COVID-19 epidemic has confirmed the danger of drug-resistant strains and made the task of developing new antiviral drugs more urgent than ever. The presented short review showed the main methodologies for the synthesis of antiviral derivatives of sulfonamides, which were obtained on the basis of *N*-containing heterocyclic structures. There is no doubt about the effect of the sulfonamide fragment on the antiviral properties of the obtained compounds. Moreover, sulfonamides can be both sources of nitrogen for building a nitrogen-containing heterocyclic core and the side chain substituents of a biologically active substance. The formation of the sulfonamide group is often achieved by the reaction of the *N*-nucleophilic center in the substrate molecule with the corresponding sulfonylchloride. This review provides examples of the synthesis of sulfonamide compounds with antiviral properties that can be used to develop drugs against EMCV, AdV5, HPIV, EBOV, MARV, APMV, SARS-CoV-2, HIV, DENV, ZIKV, BVDV, NDV, IBDV, H9N2, AIV, IBV, HSV, CBV, HAV, VYG, HCVcc and HAdV.

## Data Availability

Not applicable.

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
