# Peer review of "Sulfonamides with Heterocyclic Periphery as Antiviral Agents"

_molecules, 2022, doi:10.3390/molecules28010051_

Round 1
Reviewer 1 Report
This is potentially interesting and useful review with important topic. It can be published in Molecules.
However, I see some issues, which should be addressed before the publication.
First, the title should be more specific (e.g. “Sulfonamides with heterocyclic periphery as antiviral agents”.
Second, sulfonamides are well-known pharmacophore for many clinically validate targets, for example carbonic anhydrase. So the author should revise and re-write the introduction (and abstract). In the current form the introduction is not represents the state-of-art level of the sulfonamide-based medicinal chemistry.
I recommend familiarizing with works of Prof. Claudiu Supuran and Prof. Mikhail Krasavin. Several representative references, which can be useful for the introduction improvement: 10.1080/14756366.2021.1927007; 10.1016/j.ejmech.2019.111642; 10.3390/ijms222413405; 10.1016/j.ejmech.2015.06.022; 10.1080/14756366.2022.2056733.
Third, lines 29-31: the author should clarify the activity type and the relevant reference. For example, hypoglycemic [4], antileprotic [6] etc. Moreover, these two references should be included as relevant examples of the biological diversity of sulfonamides: 10.1016/j.bmcl.2019.126677; 10.1016/j.ejmech.2014.07.023.
Fourth, the author missed one relevant antivirus scaffold, which was investigated by several groups (10.1016/j.bioorg.2015.07.005; 10.1371/journal.pbio.3000281; 10.3390/life12111832). It must be mentioned.
Author Response
Reviewer 1
This is potentially interesting and useful review with important topic. It can be published in Molecules.
However, I see some issues, which should be addressed before the publication.
First, the title should be more specific (e.g. “Sulfonamides with heterocyclic periphery as antiviral agents”.
DONE. The title were changed.
Second, sulfonamides are well-known pharmacophore for many clinically validate targets, for example carbonic anhydrase. So the author should revise and re-write the introduction (and abstract). In the current form the introduction is not represents the state-of-art level of the sulfonamide-based medicinal chemistry.
I recommend familiarizing with works of Prof. Claudiu Supuran and Prof. Mikhail Krasavin. Several representative references, which can be useful for the introduction improvement: 10.1080/14756366.2021.1927007; 10.1016/j.ejmech.2019.111642; 10.3390/ijms222413405; 10.1016/j.ejmech.2015.06.022; 10.1080/14756366.2022.2056733.
DONE. The abstract and introduction were improved. References and comments to works of Prof. Claudiu Supuran and Prof. Mikhail Krasavin (10.1080/14756366.2021.1927007; 10.1016/j.ejmech.2019.111642; 10.3390/ijms222413405; 10.1016/j.ejmech.2015.06.022; 10.1080/14756366.2022.2056733) (ref. [22-26]) were added. Thanks the reviewer for improvement!
Third, lines 29-31: the author should clarify the activity type and the relevant reference. For example, hypoglycemic [4], antileprotic [6] etc.
DONE. Activity types were clarified, references were added. (Ref. [1-21]).
Moreover, these two references should be included as relevant examples of the biological diversity of sulfonamides: 10.1016/j.bmcl.2019.126677; 10.1016/j.ejmech.2014.07.023.
DONE. Examples of the biological diversity of sulfonamides (10.1016/j.bmcl.2019.126677; 10.1016/j.ejmech.2014.07.023) (Ref. [20-21]) were added.
Fourth, the author missed one relevant antivirus scaffold, which was investigated by several groups (10.1016/j.bioorg.2015.07.005; 10.1371/journal.pbio.3000281; 10.3390/life12111832). It must be mentioned.
DONE. References [77-79] and comments were added.
I thank the Reviewer for positive comments and careful review
Reviewer 2 Report
This short article reviews the preparations of new sulfonamide-containing compounds with notes about the viruses they are effective against. The text is reasonably well structured so as to make following the developments easy. This is accompanied by a bibliography containing 63 references. The authors have an economical style and rarely mention the names of the scientists whose work they are quoting. They just refer the reader to the bibliography. I suppose this is their style. The review really amounts to a list with few general principles noted. The type of chemistry probably makes this inevitable. In my opinion this area is very ‘hot.’ There is great interest in anti-viral reagents so the review is timely and will probably attract a good deal of attention. I recommend publication subject to some minor points.
Lines 56 onwards. These IC50 and SI values should only be reported to 1 decimal place – the large error limits make the subsequent decimal places meaningless.
Line 111: Change “functionalized to three-step reaction” to functionalized in a three-step sequence.
Line 165: Change “In rhe work” to: In the work
Line 178: Add ‘to’ i. e.: up to 6 times more
Scheme 9: Above the first arrow subscript the 3 of triethylamine.
Line 201: Change “There is sulfonamide-containing” to: There are sulfonamide-containing
Scheme 11: Compound 55 – there appears to be an NH group missing from the right-hand part of the structure.
Line 220: Omit “is” i.e. ‘In the work [51] was presented’
Line 231-232: Change “Compound 58 was further undergoes to heterocyclization” to: Compound 58 undergoes further heterocyclization
Line 261: Change “newly” to new
Scheme 15: The cycloalkyl ring is missing from the final structure 71.
Line 285: Change “heterocycles that widely in- volved in” to: heterocycles that are widely in-volved in
Carefully revise the English
Author Response
Reviewer 2
This short article reviews the preparations of new sulfonamide-containing compounds with notes about the viruses they are effective against. The text is reasonably well structured so as to make following the developments easy. This is accompanied by a bibliography containing 63 references. The authors have an economical style and rarely mention the names of the scientists whose work they are quoting. They just refer the reader to the bibliography. I suppose this is their style. The review really amounts to a list with few general principles noted. The type of chemistry probably makes this inevitable. In my opinion this area is very ‘hot.’ There is great interest in anti-viral reagents so the review is timely and will probably attract a good deal of attention. I recommend publication subject to some minor points.
Lines 56 onwards. These IC50 and SI values should only be reported to 1 decimal place – the large error limits make the subsequent decimal places meaningless.
DONE
Line 111: Change “functionalized to three-step reaction” to functionalized in a three-step sequence.
DONE. The word “reaction” were changed to “sequence”.
Line 165: Change “In rhe work” to: In the work
CHANGED
Line 178: Add ‘to’ i. e.: up to 6 times more
DONE
Scheme 9: Above the first arrow subscript the 3 of triethylamine.
DONE
Line 201: Change “There is sulfonamide-containing” to: There are sulfonamide-containing
CHANGED
Scheme 11: Compound 55 – there appears to be an NH group missing from the right-hand part of the structure.
NH-GROUP WERE ADDED
Line 220: Omit “is” i.e. ‘In the work [51] was presented’
CHANGED
Line 231-232: Change “Compound 58 was further undergoes to heterocyclization” to: Compound 58 undergoes further heterocyclization
CHANGED
Line 261: Change “newly” to new
CHANGED
Scheme 15: The cycloalkyl ring is missing from the final structure 71.
The cycloalkyl ring were added to the final structure 71.
Line 285: Change “heterocycles that widely in- volved in” to: heterocycles that are widely in-volved in
CHANGED
Carefully revise the English
English were revised as it possible
I thank the Reviewer for positive comments and careful review
Round 2
Reviewer 1 Report
page 12, line 346 "IC50" => IC50. Please correct it.